# LABEL DELAY IN CONTINUAL LEARNING

**Csaba Botos**[1][*][†]  **Wenxuan Zhang**[2][*]  **Matthias Müller**[3]  **Ser-Nam Lim**[4]

**Mohamed Elhoseiny**[2]  **Philip H.S. Torr**[1]  **Adel Bibi**[1]

[1]University of Oxford, [2]KAUST, [3]Intel Labs, [4]Meta

## ABSTRACT

Online continual learning, the process of training models on streaming data, has gained increasing attention in recent years. However, a critical aspect often overlooked is the label delay, where new data may not be labeled due to slow and costly annotation processes. We introduce a new continual learning framework with explicit modeling of the label delay between data and label streams over time steps. In each step, the framework reveals both unlabeled data from the current time step $t$ and labels delayed with $d$ steps, from the time step $t - d$. In our extensive experiments amounting to 1060 GPU days, we show that merely augmenting the computational resources is insufficient to tackle this challenge. Our findings underline a notable performance decline when solely relying on labeled data when the label delay becomes significant. More surprisingly, when using state-of-the-art SSL and TTA techniques to utilize the newer, unlabeled data, they fail to surpass the performance of a naïve method that simply trains on the delayed supervised stream. To this end, we introduce a simple, efficient baseline that rehearses from the labeled memory samples that are most similar to the new unlabeled samples. This method bridges the accuracy gap caused by label delay without significantly increasing computational complexity. We show experimentally that our method is the least affected by the label delay factor and in some cases successfully recovers the accuracy of the non-delayed counterpart. We conduct various ablations and sensitivity experiments, demonstrating the effectiveness of our approach.

## 1 INTRODUCTION

Machine learning models have become the de facto standard for a wide range of applications, including social media, finance, and healthcare. However, these models usually struggle when data is constantly changing over time in a never-ending stream, which is a common norm in real-world scenarios. This challenge continues to be an active area of research known as Continual Learning (CL). However, most CL prior art examine this problem with a presumption of the immediate availability of labels once the data is collected. This assumption can clash with the realities of practical applications.

Consider the task of monitoring recovery trends in patients after surgeries. Doctors gather health data from numerous post-operative patients regularly. However, this collective data does not immediately indicate broader recovery trends or potential common complications. To make informed determinations, several weeks of extensive checks and tests across multiple patients are needed. Only after these checks can they label the gathered data as indicating broader "recovery" or "complication" trends. However, by the time they gather data, assess it, label it, and train the model, new patient data or health scenarios may emerge. This leads to a repeating cycle: collecting data from various patients, assessing the trends, labeling the data, training the model, and then deploying it on new patients. The longer this cycle takes, the more likely it is going to affect the model's reliability, a challenge we refer to as **label delay**.

---

[*]Equal contribution.

[†]e-mail: `csbotos@robots.ox.ac.uk`

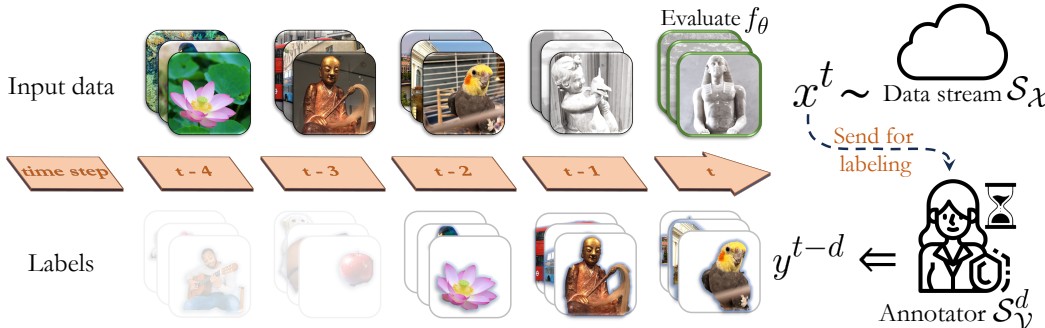

Figure 1: **Illustration of label delay.** This figure shows a typical Continual Learning setup with label delay due to annotation. At every time step $t$, the data stream $\mathcal{S}_{\mathcal{X}}$ reveals unlabeled data $x^t$, on which the model $f_\theta$ is evaluated (highlighted with green borders). Followed by that, the data is sent to the annotator $\mathcal{S}_{\mathcal{Y}}$ who takes an equivalent of $d$ time steps to provide the corresponding labels. Consequently, at time-step $t$ the label $y^{t-d}$ becomes available for the input data from $d$ time steps before. The Continual Learning model can be trained using the **delayed supervised data** (shown in color) and the **newest unsupervised data** (shown in grayscale). In this example, the stream reveals three samples at each time step and the annotation delay is $d = 2$.

In this paper, we propose a CL setting that explicitly accounts for the delay between the arrival of new data and obtaining the corresponding labels. In our proposed setting, the model is trained continually over time steps with label delay of $d$ steps. In each step, the model is revealed two sets of data one that is unlabeled from this current time step $t$, in addition to labels of the samples revealed from the step $t - d$. In the first part of our experiments, we analyze the naïve approach of simply waiting for the labels and training exclusively on supervised data. We show that this is a challenging setting where the performance of the CL model consistently degrades as the delay increases. Moreover, we find that simply increasing the number of parameter updates per time step does not resolve the problem. In the second part, we propose to utilize the unlabeled data from the most recent distribution by integrating two promising paradigms into our setting: Self-Supervised Learning (SSL) and Test-Time Adaptation (TTA). Surprisingly, out of the 6 SSL methods and 4 TTA methods considered, none could outperform the naïve baseline given the same computational budget. To this end, we propose a simple baseline that samples from previously rehearsed labeled data to match the distribution of the newest data distribution. We show that for some of the experiments, this approach successfully closes the accuracy gap caused by the label delay.

In summary, our contributions are:

- We propose a new formal Continual Learning setting that factors label delays between the arrival of new data and their corresponding labels due to the annotation latency of the annotation process.
- We conduct extensive experiments ($\sim 25,000$ of GPU hours) on various Online Continual Learning datasets, such as CLOC (Cai et al., 2021) and CGLM (Prabhu et al., 2023b). Following recent prior art on Budgeted Continual Learning (Prabhu et al., 2023a; Ghunaim et al., 2023), we compare the best performing Self-Supervised Learning (Balestriero et al., 2023) and Test Time Adaptation (Liang et al., 2023) methods and find that none of them outperforms the naïve baseline that simply ignores the label delay and trains a model on the delayed labeled stream.
- We propose **I**mportance **W**eighted **M**emory **S**ampling to rehearse past labeled data most similar to the most recent unlabeled data, bridging the gap in performance. IWMS outperforms the naïve method significantly and improves over SSL and TTA method over diverse delay and computational budget scenarios with a negligible increase in computational complexity. We further present an in-depth analysis of the proposed method.

## 2 RELATED WORK

**Continual Learning.** Early work on continual learning primarily revolved around task-based continual learning, where models adapt to new tasks as they are presented sequentially. These

approaches heavily rely on task boundaries, aiding models in recognizing and adapting to each task separately (Caccia et al., 2021; Aljundi et al., 2019a). Conversely, recent work is done in the task-free continual learning setting, where explicit task boundaries are absent, and data distributions evolve over time (Aljundi et al., 2019b;c; Cai et al., 2021). This scenario poses a challenge for models to adapt without clear task demarcations. Prabhu et al. (2020; 2023a) illustrated that minimalistic methods can outperform both offline and online continual learning approaches. This was also later observed recently by RealtimeOCL by Ghunaim et al. (2023), which reports experience replay as the most effective method when realistic computational constraints are considered and methods are normalized by their corresponding complexities. An essential distinction between the complexity-normalized framework introduced in RealtimeOCL and our delayed label setting lies in the concept of *delay*. In RealtimeOCL, delay arises from model complexity: in their *fast-stream* scenario, the stream releases input-label pairs quicker than models can update, causing models to train on a belated batch of samples. In essence, while in Ghunaim et al. (2023) the delay emerges from model evaluation constraints, labels are still instantly available. In contrast to both RealtimeOCL and prior online continual learning approaches, our work examines delays attributed to the non-instantaneous arrival of labels , while under normalized computational budget. The delay in our setup creates a distribution discrepancy between training and evaluation samples due to the time-varying stream distribution. Hammoud et al. (2023) highlighted the exploitation of label-correlation in online continual learning methods, with a focus on improving online accuracy metrics through future samples evaluation. In contrast, our work models the real-world annotation time cost and leverages unlabeled data to enhance performance on the standard evaluation metric.

**Self-Supervised Learning.** There is a rich literature of self-supervised learning (SSL) over the recent years. Early works such as MOCO (He et al., 2020; Chen et al., 2021) and SimCLR (Chen et al., 2020) focused on differentiating between positive and negative examples through maximizing similarity among positive pairs and minimizing it among negative pairs. BYOL (Grill et al., 2020) and SimSiam (Chen & He, 2021) further investigate the necessity of negative sampling and explore using a copy of the model for target representations instead. Barlow Twins (Zbontar et al., 2021) and VICReg (Bardes et al., 2022) encourage similarity between distorted versions of an example as alternatives of contrastive loss. As such, a growing line of work adapts SSL to continual learning to make use of unlabeled data, such as CaSSLe (Fini et al., 2022) in task-agnostic setting and SCALE (Yu et al., 2023) in task-free setting. However, most previous work did not perform comprehensive examination of the abovementioned three typical categories of SSL methods (Balestriero et al., 2023). We will bridge this gap in our work.

**Test Time Adaptation.** While SSL methods are mainly used as computationally extensive pre-training techniques, TTA methods are designed to adjust models efficiently from their original training distribution to the distribution from which the evaluation samples are drawn from. Given our problem formulation, TTA methods appear suitable; hence, it is crucial to assess the efficacy of these methods within our framework. Entropy regularization methods like SHOT (Liang et al., 2020) and TENT (Wang et al., 2021) update the feature extractor or learnable parameters of the batch-normalization layers (Ioffe & Szegedy, 2015) to minimize the entropy of the predictions. Nguyen et al. (2023) introduced TIPI, a method that finds input transformations that can simulate the domain shifts to enforce the model to be robust to the distribution shifts. SAR (Niu et al., 2023) incorporates an active sampling scheme to filter samples with noisy gradients. More recent works consider Test Time Adaptation in online setting (Alfarra et al., 2023) or Continual Learning setting (Wang et al., 2022). In this work, we extend this by applying test time adaptation in scenarios where the distribution of future test samples is unknown and cannot be assumed to be stationary.

## 3  PROBLEM FORMULATION

We follow the conventional online continual learning problem definition proposed by Cai et al. (2021). In such a setting, we seek to learn a model $f_\theta : \mathcal{X} \to \mathcal{Y}$ on a stream $\mathcal{S}$ where for each time step $t \in \{1, 2, \dots \}$ the stream $\mathcal{S}$ reveals data from a time-varying distribution $\mathcal{D}_t$ sequentially in batches of size $n$. At every time step, $f_\theta$ is required to predict the labels of the coming batch $\{x_i^t\}_{i=1}^n$ first. Followed by this, the corresponding labels $\{y_i^t\}_{i=1}^n$ are immediately revealed by the stream. Finally, the model is updated using the most recent training data $\{(x_i^t, y_i^t)\}_{i=1}^n$.

This setting, however, assumes that the annotation process is instantaneous, *i.e.* the time it takes to provide the ground truth for the input samples is negligible. In practice, this assumption rarely

holds. It is often the case that the rate at which data is revealed from the stream $\mathcal{S}$ is faster than the rate at which labels for the unlabeled data can be collected, as opposed to it being instantaneously revealed. To account for this delay in accumulating the labels, we propose a setting that accommodates this lag in label availability while still allowing for the model to be updated with the most recent unlabeled data.

We modify the previous setting in which labels of the data revealed at time step $t$ will only be revealed after $d$ time steps in the future. Or equivalently, the stream $\mathcal{S}$ can be decoupled into two streams, one revealing the data $\mathcal{S}_{\mathcal{X}}$ and the other revealing the labels $\mathcal{S}_{\mathcal{Y}}$, where the amount of delay between the two streams is effectively the relative speed between the rate at which data is revealed and at which it can be annotated. Consequently, at every time step, the stream reveals the labels for the samples from $d$ time steps before $\{(x_i^{t-d}, y_i^{t-d})\}_{i=1}^{n}$ and the input data of the current time step $\{x_i^t\}_{i=1}^{n}$.

Following recent prior art (Prabhu et al., 2020; 2023a) that argues for normalized computation for fair comparisons, the models are given a fixed computational budget $\mathcal{C}$ to update the model parameters from $\theta_t$ to $\theta_{t+1}$ for every time step $t$. To that end, our new proposed setting can be formalized, alternatively to the classical OCL setting, as follows: per time step $t$,

1. The stream $\mathcal{S}_{\mathcal{X}}$ reveals a batch of images $\{x_i^t\}_{i=1}^n \sim \mathcal{D}_t$;
2. The model $f_{\theta_t}$ makes predictions $\{\hat{y}_i^t\}_{i=1}^n$ for the new revealed batch $\{x_i^t\}_{i=1}^n$;
3. The stream $\mathcal{S}_{\mathcal{Y}}$ reveals true labels $\{y_i^{t-d}\}_{i=1}^n$;
4. The model $f_{\theta_t}$ is evaluated by comparing the predictions $\{\hat{y}_i^t\}_{i=1}^n$ and true labels $\{y_i^t\}_{i=1}^n$, where the true labels are only for testing;
5. The model $f_{\theta_t}$ is updated to $f_{\theta_{t+1}}$ using labeled data $\cup_{\tau=1}^{t-d}\{(x_i^\tau, y_i^\tau)\}_{i=1}^n$ and unlabeled data $\cup_{\tau=t-d}^{t}\{x_i^\tau\}_{i=1}^n$ under a computational budget $\mathcal{C}$.

Note that this means at each time step $t$, the stream reveals a batch of *non-corresponding* images $\{x_i^t\}_{i=1}^n$ and labels $\{y_i^{t-d}\}_{i=1}^n$, as illustrated in Figure 1. With the label delay of $d$ time steps, the images themselves revealed from time step $t-d$ to time step $t$ can be used for training, despite that labels are not available. A naïve way to solve this problem is to discard the unlabeled images and only train on labeled data $\cup_{\tau=1}^{t-d}\{(x_i^\tau, y_i^\tau)\}_{i=1}^n$. However, note that the evaluation is still done on the most recent samples, from $\mathcal{S}_{\mathcal{X}}$. This effectively means, that the model, while being evaluated on the newest data, is trained on older data (in particular, at least $d$ time steps older). Since in our setting, the distribution from which the training and evaluation samples are drawn from is not stationary, this discrepancy is highly likely to hinder the model's performance.

## 4 PRACTICAL IMPLICATIONS OF LABEL DELAY

We first introduce our experimental setup in Section 4.1. Then, we demonstrate in Section 4.2 that the label delay in an online continual learning setting poses a difficult challenge to continual learning algorithms by showing that performance degrades as the label delay increases. Moreover, we investigate how much of the performance gap can be bridged by increasing the computational budget $\mathcal{C}$ in Section 4.3.

### 4.1 EXPERIMENTAL SETUP

**Datasets.** We conduct our experiments on two large-scale online continual learning datasets. The first, Continual Localization (CLOC) (Cai et al., 2021) which contains 39M images from 712 geolocation ranging from 2007 to 2014. The second is Continual Google Landmarks (CGLM) (Prabhu et al., 2023b) which contains 430K images over 10788 classes. The task is to localize the input image to one of the 712/10788 locations, respectively, for CLOC and CGLM. We follow the same split as in the common use of CLOC (Cai et al., 2021) and CGLM (Prabhu et al., 2023b) and report the *Online Accuracy* metric detailed in (Cai et al., 2021)

**Architecture and Optimization.** We use ResNet18-BN (He et al., 2016) for backbone architecture. Similarly to (Ghunaim et al., 2023; Prabhu et al., 2023a), in our experiments, the stream reveals a mini-batch, with the size of $n = 128$ for CLOC and $n = 64$ for CGLM. We use SGD with the learning rate of 0.005, momentum of 0.9, and weight decay of $10^{-5}$. We apply random cropping and resizing to the images, such that the resulting input has a resolution of $224 \times 224$. In our experiments,

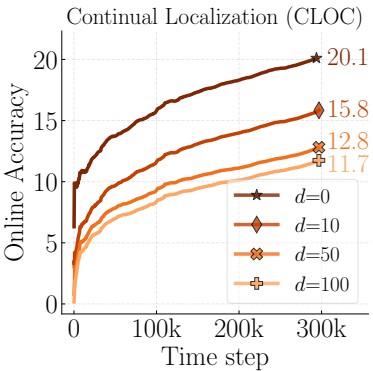 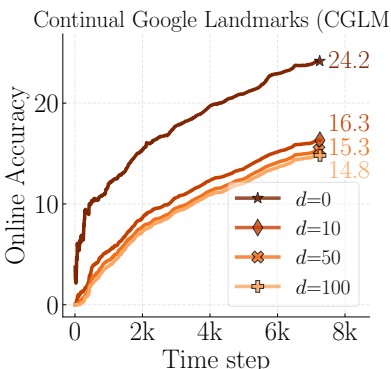

Figure 2: **Effects of Varying Label Delay.** The performance of a *Naïve* Online Continual Learner model gradually degrades with increasing values of delay $d$.

we refer to the *Naïve* method, which discards the unlabeled samples and uses uniformly sampled images from a fixed sized memory replay buffer. The memory buffer size is consistently $2^{19}$ samples throughout our experiments unless stated otherwise. The First-In-First-Out mechanism to add and discard samples to and from the buffer follows Cai et al. (2021) throughout all of our experiments. On the CLOC dataset, during a single forward and backward pass, the naive model processes the 128 most recent labeled samples and 128 random samples from the replay buffer, thus the effective batch size is 256. Due to the difference in size of the two datasets, on CGLM we follow the same procedure with *half* of the number of samples, thus resulting in an effective training batch size of 128. As our primary performance metric, we report the accuracy computed at each time step as per Step 4 from Section 3 which is referred to as Online Accuracy (Cai et al., 2021).

**Computational Budget.** Prabhu et al. (2023a) and Ghunaim et al. (2023) argue that normalizing the computational budget is necessary for fair comparison across CL methods; thus, we follow this common practice and normalize the number of FLOPs required to make one backward pass with a ResNet18 (He et al., 2016) to $\mathcal{C} = 1$. When performing experiments with a larger computational budget, we take integer multiplies of $\mathcal{C}$, and in such scenarios we apply multiple parameter update steps per stream time-steps. The proposed label delay factor $d$ represents the amount of time-steps the labels are delayed with. Note that, for $\mathcal{C} = 1, d = 0$, our experimental setting is identical to that of Cai et al. (2021) and Ghunaim et al. (2023).

## 4.2 ACCURACY DEGRADATION CAUSED BY LABEL DELAY

In Figure 2, we analyze how varying the label delay $d \in \{0, 10, 50, 100\}$ impacts the performance of the Naïve continual learning model on both CLOC and CGLM. We show that the delay factor does not cause immediate performance degradation. Moreover, there is a clear correspondence between $d$ and the extent of accuracy drop.

More specifically, we see that in both datasets the final accuracy score gradually drops from 14.8% to 12.0%, 10.0%, and 9.1% over $d = 10, 50, 100$, respectively, leading to more than 38% drop in performance in the worst-case scenario on CLOC. Similarly, on CGLM increasing the label delay from $d = 10$ to 50 and 100 leads to consistently decreasing accuracy 13.4%, 12.5% and 12.0% respectively. However, in comparison to CLOC, on CGLM we observe a more rapid decrease in performance from even the smallest delay $d = 10$ leading to a 25% relative accuracy drop, while in the largest label delay scenario we report a 32% relative accuracy gap between the delayed and the non-delayed model.

We note that the performance of the model is relatively low, and we argue that it is because the models illustrated in Figure 2 are trained under the hard computational constraints ($\mathcal{C} = 1$ in CLOC and $\mathcal{C} = 4$ in CGLM) , where only a limited number of parameter updates takes place between each time step. Our hypothesis is that the performance degradation is caused by the label delay due to the induced distribution shift between the evaluation and the training data. In the next section, we study whether the performance drop can be resolved by simply increasing the computational budget.

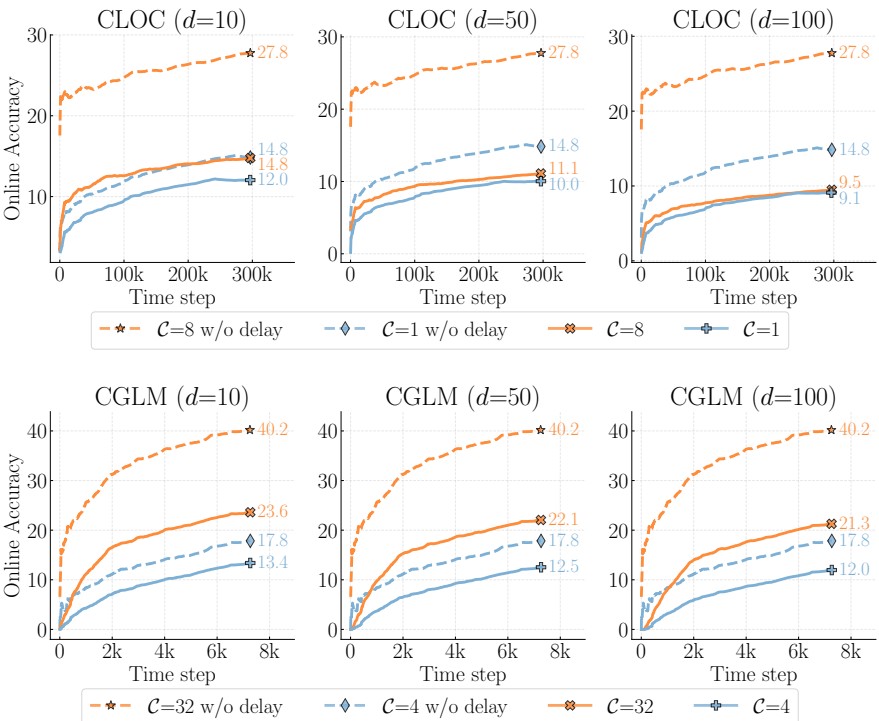

Figure 3: **Increasing the Computational Budget $\mathcal{C}$ under varying Label Delay $d$ settings.** We report our scores on two datasets, CLOC (top) and CGLM (bottom) across three delay scenarios $d = 10, 50, 100$ on the left, middle, and right respectively. In the easiest setting, $d = 10$ the accuracy gap can be bridged by simply spending more $8\times$ more compute. In more challenging scenarios, $d = 50, 100$ naively increasing the number of updates shows diminishing returns, motivating us to find better use of the increased computational budget. For brevity, we refer to the non-delayed $d = 0$ Naïve method as "w/o delay".

### 4.3 CAN WE BRIDGE THE GAP WITH INCREASED $\mathcal{C}$?

Despite not being always feasible, we examine one potential solution to recover the performance of the Naïve Baseline when $d > 0$, which is increasing the computational budget $\mathcal{C}$ and performing multiple parameter updates per time step. In Figure 3, we show results when the computational budget of the Naïve is doubled in three steps, *i.e.*, $\mathcal{C} = 2, 4, 8$ under three delays of $d \in \{10, 50, 100\}$. In Figure 3, we observe the performance improvement of the Naïve method is proportional to the budget $\mathcal{C}$ under all delays. For example, when $d = 10$, the model can recover the performance of the Naïve accuracy in the $d = 0$ setting by increasing the computational budget from $1 \rightarrow 8$ and $4 \rightarrow 8$, on CLOC and CGLM, respectively. On the CLOC dataset (top row of Figure 3) where the label delay is significant ($d = 50, 100$), the Naïve does not reach the performance of its non-delayed counterpart, even when the computational budget is increased from $\mathcal{C} = 1 \rightarrow 8$ (solid orange curve). On the other hand, increasing from $\mathcal{C} = 4 \rightarrow 32$ on CGLM (solid orange curve in the bottom row of Figure 3) results in the delayed Naïve consistently outperforming its non-delayed counterpart with the original budget $\mathcal{C} = 4$ (solid blue curve). However, when compared against the non-delayed Naïve method with matched computational budget (dashed orange curve), the delay causes increased accuracy gaps: $-16.6\%, -18.1\%$ and $-18.9\%$, with respect to the delay factor $d = 10, 50, 100$.

**Section Conclusion.** To facilitate a comprehensive understanding of the implications, we compared the Naïve method with increased $\mathcal{C}$ against its non-delayed counterpart, under both the original and matched computational budgets. Under matched computational budget, the performance is still significantly worse, motivating us to find methods that can leverage the more recent unlabeled data to bridge the gap. We have shown that increasing the computational budget of an Online Continual Learning model alleviates the problem of the label delay, but that such a solution not only faces diminishing returns but becomes less effective under larger delays. This motivates our next set of

experiments, in which we augment the Naïve model training by utilizing the input images *before* their corresponding labels become available.

## 5 Utilizing Data Prior to Label Arrival

In our proposed online continual learning with label delay, as observed earlier, the larger the delay the more challenging it is for Naïve, a method that relies only on older labeled data, to effectively predict new samples. This is due to a larger gap in distribution between the samples the model is trained on and the ones the model is evaluated on. The drop in accuracy caused by this discrepancy can be partially resolved by training the models longer, nevertheless this approach is not satisfactory for multiple reasons. First, in many real-world scenarios, the computational budget is fixed. Furthermore, even if it can be increased, we showed that, due to the diminishing returns, the accuracy gap remains substantial. This begs the question of whether the new unlabeled data can be used for training to improve over Naïve, as it is much more similar to the data that the model is evaluated on.

In our experiments, we explore the techniques of the two most promising paradigms for utilizing the unlabeled data, namely, Self-Supervised Learning (SSL) and Test Time Adaptation (TTA). We integrate several of each family of methods into our setting and evaluate them under various delays and computational budgets. In particular, we augment the Naïve method by optimizing the supervised objective $\mathcal{L}_S$ (which is the standard Cross Entropy loss over the labeled data in our case) and an unsupervised regularization term $\mathcal{L}_U$ provided by the underlying methods.

### 5.1 Self-Supervised Learning to handle label delay?

**Experimental Setup.** When it comes to integrating SSL methods in our setting, there are multiple possible ways to utilize the learning methods, leading to vastly different results. We find the overall best performing variant iteratively optimizes $\mathcal{L}_S$ and $\mathcal{L}_U$. More specifically, we optimize $\mathcal{L}_S$ on the labeled data (identically to naive) and optimize the contrastive loss $\mathcal{L}_U$ on the unlabeled data in an alternating fashion, until the computational budget $\mathcal{C}$ is not consumed. We experiment with the three main families of SSL, *i.e.*, Deep Metric Learning Family (MoCo (He et al., 2020), SimCLR (Chen et al., 2020),and NNCLR (Dwibedi et al., 2021)), Self-Distillation (BYOL (Grill et al., 2020) and SimSIAM (Chen & He, 2021), and DINO (Caron et al., 2021)), and Canonical Correlation Analysis (VICReg (Bardes et al., 2022), BarlowTwins (Zbontar et al., 2021), SWAV (Caron et al., 2020), and W-MSE (Ermolov et al., 2021)). Furthermore, we conduct hyperparameter tuning on the first 10K iterations of the training data over all methods, including a scalar constant $\lambda_U$ that rescales $\mathcal{L}_U$.

**Computational Budget.** For fair comparison, and following Prabhu et al. (2023a); Ghunaim et al. (2023), we normalise the computational complexity of the compared methods. We find that while SSL methods may take multiple forward passes, potentially with varying input sizes, the backward pass is consistently done only once among the variants; therefore we choose the number of backward passes to measure the computational complexity of the resulting methods. According to this computational complexity metric, the Naïve method augmented with SSL at each time step takes two backward passes, one for computing the gradients of $\mathcal{L}_U$ and one for $\mathcal{L}_S$, thus $\mathcal{C}_{\text{SSL}} = 2$.

**Observations.** In Figure 4, we report the results of the best of ten SSL methods on both CLOC (top row) and CGLM (bottom row). We compare the SSL based approaches against Naïve under varying computational budget and label delay scenarios. As discussed earlier, the SSL based methods have an effective computational cost of $\mathcal{C}_{\text{SSL}} = 2$. To maintain fair comparison, we report the Naïve variant with an increased computational budget $\mathcal{C}_{\text{Naïve}} = 2$, *i.e.* performing twice as many supervised parameter updates. Under such settings, the best performing SSL based method, ReSSL (Zheng et al., 2021) (see the Supplementary Material A.1 for an in-depth comparison) underperforms Naïve (on CLOC) $-1.2\%, -0.2\%$ and $-0.1\%$ over $d = 10, 50, 100$, respectively. Similarly, on CGLM, for all delay scenarios, the best performing SSL method, underperforms the Naïve counterpart: $-2.3\%, -2\%$ and $-2.1\%$ in terms of online accuracy. We hypothesize this is due to several reasons: (1) SSL methods generally are sample inefficient and require a few magnitudes more iterations than the supervised method to achieve the same downstream accuracy. (2) The contrastive objective aids learning low level visual concepts, features that are less important for the downstream task of CLOC and CGLM, *i.e.* geo-localization and landmark detection respectively.

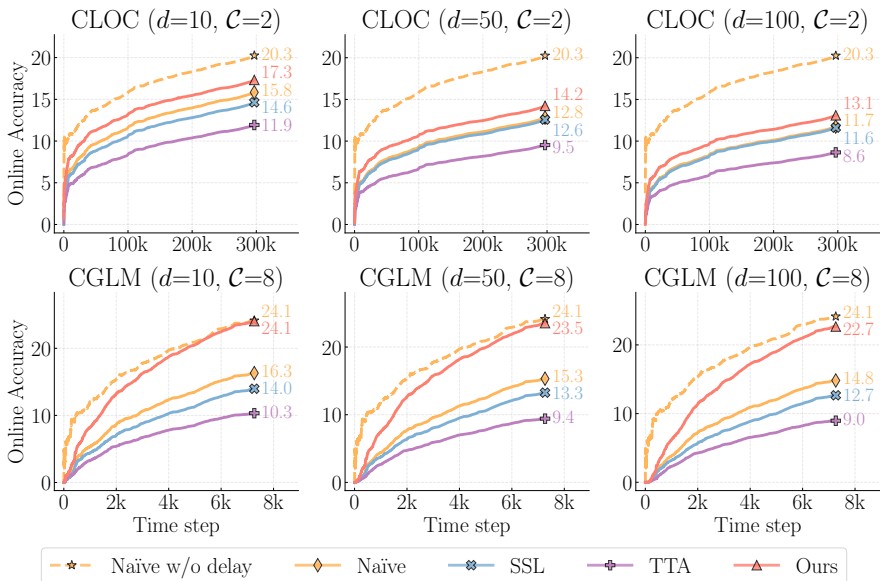

Figure 4: **Comparison of various unsupervised methods.** The performance of the Naïve model augmented with various unsupervised adaptation methods under varying label delay $d$ settings on two datasets (top: CLOC (Cai et al., 2021), bottom: CGLM (Prabhu et al., 2023a)). We report the best performing methods in each category, ReSSL (Chen et al., 2021) and CoTTA (Wang et al., 2022) for SSL and TTA respectively. Our proposed method, described in Section 5.3, consistently outperforms all categories under all delay settings on both datasets.

## 5.2 TEST-TIME ADAPTATION

**Experimental Setup.** Similarly to SSL, there are multiple ways to integrate TTA methods in our setting. We find the following setup resulting in the best performance: we continuously train the model with the supervised data as in the Naïve method, and before each evaluation step we adapt the model using a single parameter update on the unsupervised data $\{x_i^t\}_{i=1}^n$. We implement and compare the following TTA methods: TENT (Wang et al., 2021), EATA (Niu et al., 2022), SAR (Niu et al., 2023), and CoTTA (Wang et al., 2022), in Figure 4. For an extensive comparison of the methods under various delay scenarios, see Section A.2.

**Computational Budget.** Similarly to SSL based approaches, we normalise the computational budget for all TTA approaches by counting the full backward passes made on the unlabeled samples when optimizing $\mathcal{L}_U$. Following this complexity metric, the implemented TTA methods fall under the same associated cost; therefore, we can use the same normalization factor for all TTA methods across our experiments.

**Observations.** In Figure 4, we find that the best performing TTA method, CoTTA (Wang et al., 2022), consistently underperforms both the Naïve and the SSL baseline under every delay scenario on both datasets, from $3.1\%$ up to $-6.3\%$ performance drop. We hypothesize that the TTA methods fail to outperform the Naïve counterpart because the common assumptions among the settings on which TTA methods are evaluated are broken. The common assumptions of TTA methods are: (1) before the adaptation takes place, the model has already converged and achieved a good performance on the training data, (2) the test data distribution is stationary and a sufficient amount of unsupervised data is available for adaptation. In contrast, in our setting the source model is continuously updated between time steps and only a limited number of samples are available from the newest distribution for adaptation.

## 5.3 REHEARSING ON RELEVANT SAMPLES

When utilizing unsupervised samples to enhance the Naïve method, we have shown that directly spending the computational budget to optimize a generic objective over the unlabeled images does not seem to help to outperform the Naïve method. This suggests a solution that leverages supervised samples directly, rather than spending the computational budget on unlabeled data. Although we

cannot directly adapt the model to the newest distribution of the stream, we can adapt the sampling process from the memory buffer to match the desired distribution. We propose a very simple, yet effective method to achieve a data efficient and therefore computationally efficient solution to the challenges imposed by label delay that we dub **Importance Weighted Memory Sampling (IWMS)**.

When sampling the supervised training data, using the most recent labeled samples leads to fitting the model to an outdated distribution (we further discuss this in 5.4). Thus, we replace the newest supervised data by a batch which we sample from the memory buffer, such that the distribution of the selected samples matches the newest unlabeled data distribution. More specifically, our sampling process consists of three stages. First, at each time step $t$, for every unsupervised sample $x_j^t$ in the batch of size $n$, we compute the prediction $\tilde{y}_j^t$, and select every labeled sample from the memory buffer $(x_i^M, y_i^M)$ such that the true label of the selected samples matches the predicted label $y_i^M = \tilde{y}_j^t$. Then, we compute the pairwise cosine feature similarities $K_{i,j}$ between the unlabeled sample $x_j^t$ and the selected labeled samples $x_i^M$ by $K_{i,j} = \cos\left(h(x_i^M), h(x_j^t)\right)$, where $h$ represents the feature extractor directly before the final classification layer. Lastly, we select the most relevant supervised samples $(x_{i'}^M, y_{i'}^M)$ by sampling $i' \in \{1 \ldots |M|\}$ from a multinomial distribution with parameters $K_{:,j}$. Thus, we rehearse samples from the memory which (1) share the same true labels as the predicted labels of the unlabeled samples, (2) have high feature similarity with the unlabeled samples.

**Implementation Details.** To avoid re-computing the feature representation $h$ for each sample in the memory buffer at every iteration, during the evaluation phase, we store each input sample and their corresponding features in the memory buffer. We only compute the features once, when adding them to the replay buffer.

**Computational Budget.** Since our method simply replaces the newest supervised samples with the most similar samples from the replay buffer, we do not require any additional backward passes to compute the auxiliary objective. Therefore, the computational budget of our method is identical to the Naïve baseline, *i.e.*, $\mathcal{C}_{\text{Ours}} = 1$.

**Observations.** First, in Figure 4 on CLOC (top row), we show that our method is the only method that can outperform the Naïve baseline, by $+1.5\%, +1.4\%, +1.4\%$ improvement on $d = 10, 50, 100$ settings respectively. Although our method outperforms every delayed method, there is still a considerable gap between the non-delayed Naïve counterpart, on which we provide further explanation in the Supplementary Material A.4. In contrast, on CGLM (Fgiure 4 bottom row) we show that our method can close gap between the *non-delayed* Naïve method under all delay settings ($d = 10, 50, 100$, shown in increasing order from left to right), thus the performance drop due to the label delay is minimized to $-0\%, -0.6\%$ and $-1.4\%$ respectively.

## 5.4    ANALYSIS OF IMPORTANCE WEIGHTED MEMORY SAMPLING

In this section, we first perform an ablation study of our IWMS to show the effectiveness of the importance sampling. Then, we show our performances under different computational budgets and buffer sizes.

**Analysis on Memory Sampling Strategies.** During training, the Naïve method uses the most recent labeled data and a randomly sampled mini-batch from the memory buffer for each parameter update. Our method provides a third option for constructing the training mini-batch, which picks the labeled memory sample that is most similar to the unlabeled data. When comparing sampling strategies, we refer to the newest batch of data as (N), the random batch of data as (R) and the importance weighted memory samples as (W). In Figure 5 left, we first show that in both delay scenarios ($d = 10$ and $d = 100$) replacing the newest batch (N) with (W) results in almost doubling the performance: $+8.5\%$ and $+9.1\%$ improvement over Naïve, respectively. Interestingly enough, when we replace the (N) with uniformly sampled random buffer data (R) we report a significant increase in performance. We attribute this phenomenon to the detrimental effects of label delay: even though Naïve uses the most recent supervised samples for training, the increasing discrepancy caused by the delay $d = 10$ and $d = 100$ forces the model to over-fit on the outdated distribution.

**Analysis on Computational Budget.** We study our algorithm under various computational budgets, ranging from $\mathcal{C} = 4$ to $\mathcal{C} = 32$ (shown in Figure 5 middle). Under both delays, the performances of our algorithm improve as the computational budget improves. However, our required computational budget for the algorithm to converge is relatively small. When we increase the budget from $\mathcal{C} = 8$ to

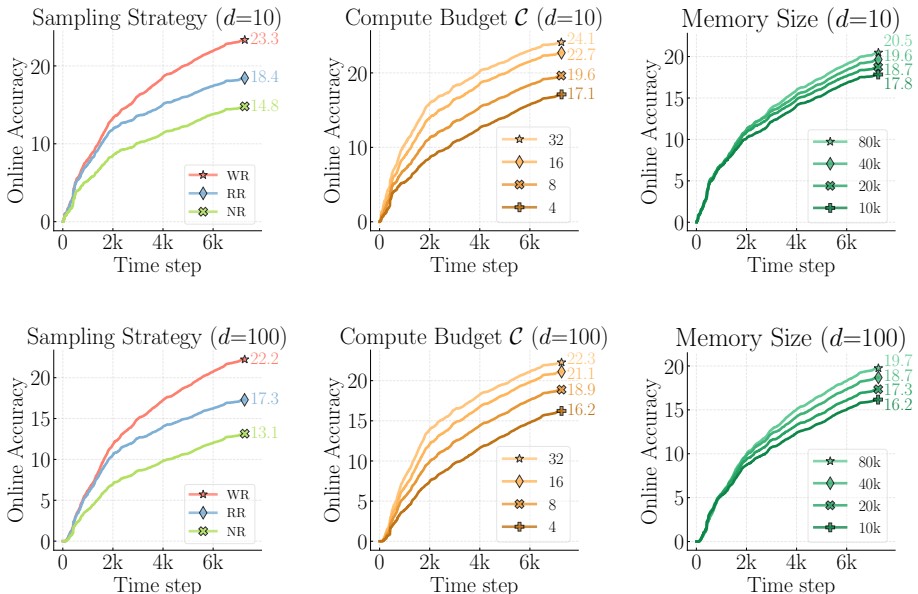

Figure 5: **Effect of sampling strategy (left), memory sizes (middle) and computational budget (right).** We report the Online Accuracy under the least (top: $d = 10$) and the most challenging (bottom: $d = 100$) label delay scenarios on CGLM (Prabhu et al., 2023b).

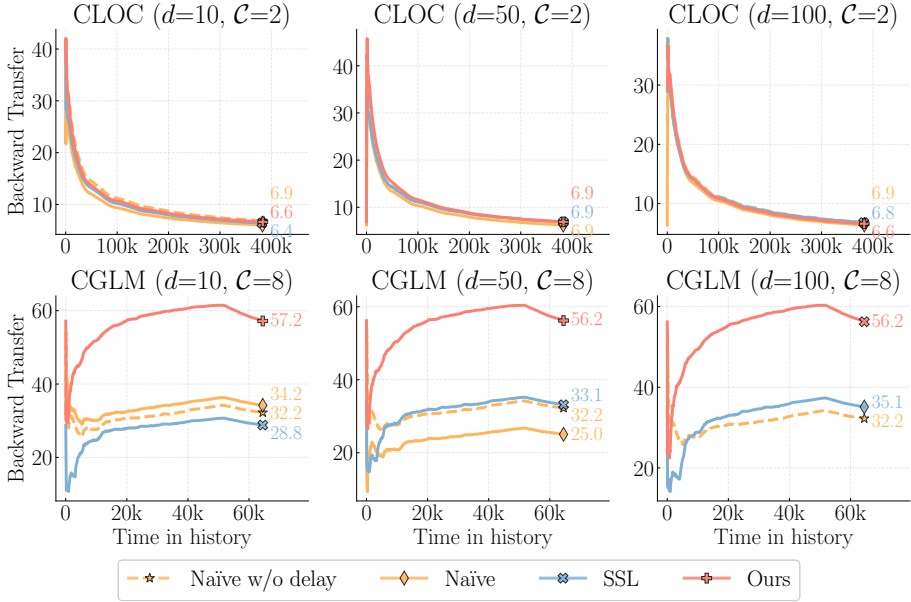

Figure 6: ADDED DURING ICLR REBUTTAL: Backward transfer for measuring forgetting on a withheld dataset.

$\mathcal{C} = 32$, the performances only increase around $+3\%$. Thus, our algorithm can efficiently cope with continual learning settings where the budget is limited.

**Analysis on the Memory Size.** We study the influence of buffer size on our proposed IWMS. In particular, we show the performance of our algorithm under the buffer size from 10k to 80k in Figure 5 (right). Even though IWMS relies on the images sampled from the buffer to represent the new coming distribution, its performances remain robust under different buffer sizes. Under both delay

of 10 and 100, the performance gap of our algorithm between the buffer size of 10k and 80k is only around 2.5%. This means our algorithm is flexible in coping with various buffer assumption.

**Analysis on forgetting over past samples** ADDED DURING ICLR REBUTTAL: In Figure 6, we observe that our method outperforms the SSL, Naïve and non-delayed Naïve baseline achieving 2x better accuracy on CGLM, whereas on CLOC all methods perform similarly (due to poor data quality as reported in the supplementary material Sec A.4).

## 6 DISCUSSION

In this work, we have addressed label delay in online continual learning. We show that merely augmenting the computational resources is insufficient to tackle this challenge. Our findings underline a notable performance decline when solely relying on labeled data when the label delay becomes significant. Moreover, state-of-the-art SSL and TTA techniques fail to surpass the performance of the naïve method of simply training on the delayed supervised stream. To address these challenges, we propose IWMS, which not only mitigates the accuracy discrepancies due to label delay, but also significantly exceeds the performance of its non-delayed counterpart, particularly in extreme delay and computational constraint conditions. Although our continual learning framework assumes a constant label delay factor and eventually provides labels for all samples revealed by the stream, in more complex applications the labeling process may exhibit variable durations and the samples selected for labeling can be guided. By looking into these scenarios, we might find ways to use Active Learning to improve sample selection in Continual Learning, pointing to a new research direction.

## 7 ACKNOWLEDGEMENT

This work is supported by a UKRI grant Turing AI Fellowship (EP/W002981/1) and EPSRC/MURI grant: EP/N019474/1. Adel Bibi has received funding from the Amazon Research Awards. The authors thank Razvan Pascanu and João Henriques for their insightful feedback. We also thank the Royal Academy of Engineering.

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

# A SUPPLEMENTARY MATERIAL

## A.1 BREAKDOWN OF SSL METHODS

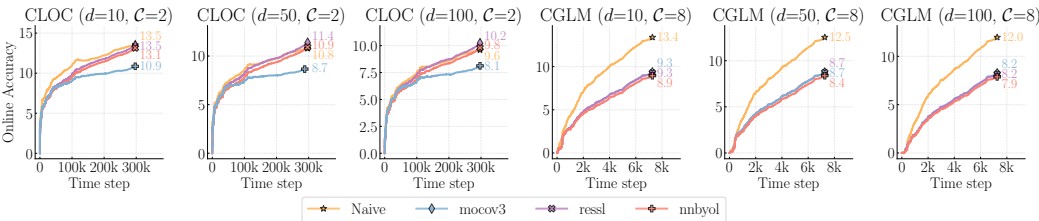

Figure 7: Comparison of the best performing SSL based methods after hyper-parameter tuning

In Figure 7 we show the performance of the best performing SSL based methods after hyper-parameter tuning. We observe that the performance of the SSL methods is highly dependent on the dataset and the delay setting. However, we apart from MoCo v3 (Chen et al., 2021), the methods perform similarly to Naïve on CLOC. On the other hand on CGLM they have insignificant differences in performance, but consistently underperform Naïve.

## A.2 BREAKDOWN OF TTA METHODS

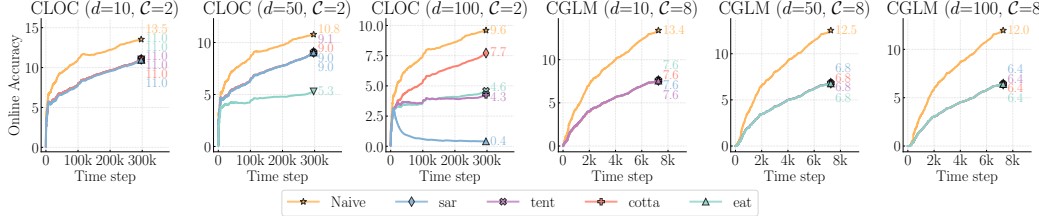

Figure 8: Comparison of the best performing TTA based methods after hyper-parameter tuning

In Figure 8 we show the performance of the best performing TTA based methods after hyper-parameter tuning. We observe that the performance of the TTA methods are consistently worse than Naïve on both CLOC and CGLM, under all delay settings. We observe that in the most severe delay scenario ($d = 100$) the performance of EAT (Niu et al., 2022) and SAR (Niu et al., 2023) is comparable to Naïve on CLOC, while CoTTA (Wang et al., 2022) avoids the catastrophic performance drop.

## A.3 COMPARISON OF SSL BASED METHODS TO NAÏVE WHEN USING SAME AMOUNT OF SUPERVISED DATA

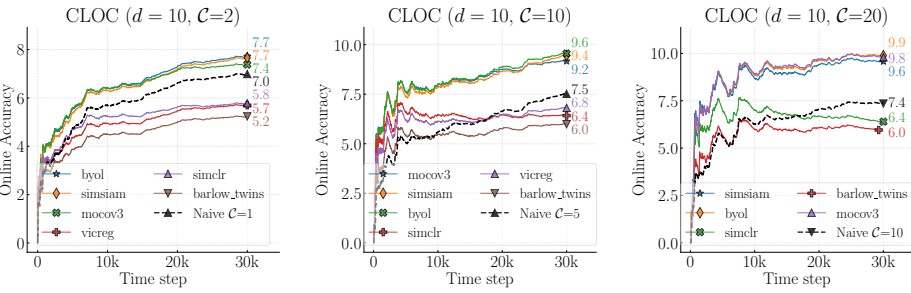

Figure 9: Detailed breakdown of various SSL methods from each family. Results are shown across varying number of parameter updates $\mathcal{C} = 2, 10, 20$ under the $d = 10$ scenario.

In Figure 9, we show that when trained on equal amount of supervised data, SSL based methods perform outperform Naïve, however the performance gap is not as significant as in the case of using the same computational budget, as shown in the main Figure 4.

## A.4 Examples of the Importance Weighted Memory Sampling on CLOC

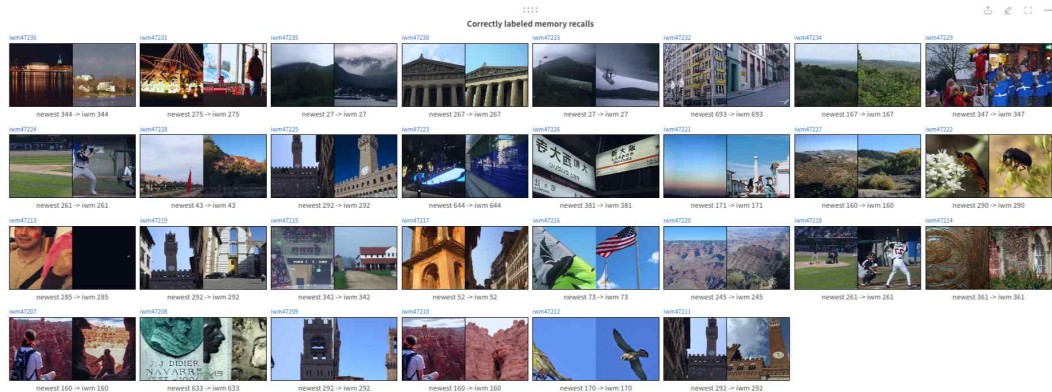

Figure 10: **Correctly labeled memory recalls.** In the subfigure's caption "Newest" refers to the newest unsupervised image observed by the model and "iwm" refers to the sample drawn from the memory by our proposed sampling method. The numbers refer to the corresponding true label IDs.

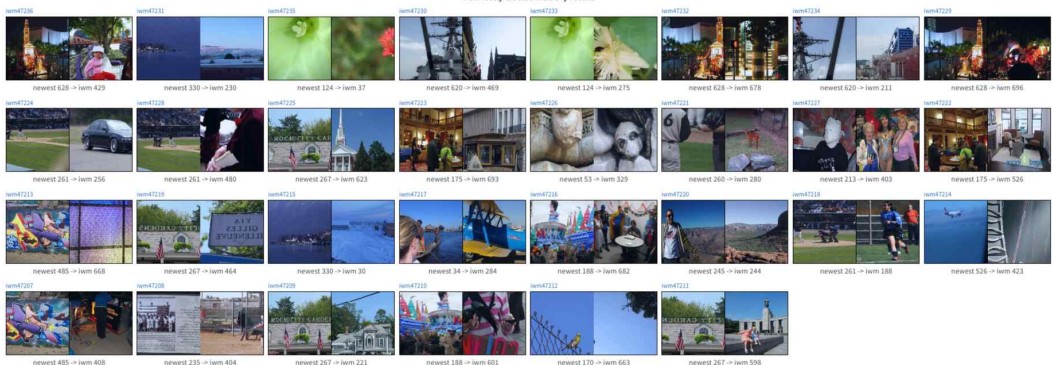

Figure 11: **Incorrectly labeled memory recalls.** In the subfigure's caption "Newest" refers to the newest unsupervised image observed by the model and "iwm" refers to the sample drawn from the memory by our proposed sampling method. The numbers refer to the corresponding true label IDs.

On CLOC, we report similar scores to Naïve due to high noise in the data. To provide evidence for our claims we visualize the supervised data sampled from the memory buffer by our Importance Weighted Memory Sampling method. In Figure 10, we show that our method is capable of guessing the correct location of the unsupervised sample (the left hand side of the image pairs) and recalling a relevant sample from memory. In contrast, the incorrect memory recalls hurt the performance even though the content of the samples might match. We illustrate such cases in Figure 11, where it is obvious that in some cases the underlying image content has no information related to the location where the picture was taken at. In such scenarios, the only way a classifier can correctly predict the labels is by exploiting label correlations, *e.g.*classifying all close-up images of flowers to belong to the same geo-location, even though the flowers are not unique to the location itself. Or consider the pictures taken at social gatherings (second row, second column from the right), where a delayed classifier without being exposed to that specific series of images has no reason to correctly predict the location ID. Our claims are reinforced by the findings of Hammoud et al. (2023).

