# OpenReview forum: "Modeling Annotation Delay In Continual Learning"
_ICLR.cc/2024/Conference — Submitted to ICLR 2024_

### Official Review · Reviewer_Leqv · 2023-10-22

**Soundness:** 2 fair
**Presentation:** 2 fair
**Contribution:** 2 fair
**Rating:** 3
**Confidence:** 2

**Summary:**

The paper studies and proposes techniques for online (continual) learning with label delay. First the paper shows that several simple techniques adapted to the considered setting provide poor performance and then, they propose a method based on sampling previous data that achieves better results.

**Strengths:**

The research topic is very relevant since the distribution of data often changes over time in practical scenarios. The paper also offers an experimental evaluation of several alternatives that can be of interest

**Weaknesses:**

The paper contribution over the state-of-the-art is unclear. Firstly, the setting the authors refer as "continual learning" is basically that of "online learning" with time-varying distributions. In particular, the model obtained at time t is never used to predict data from previous times, and hence is not affected by the specific problems of "continual learning" such as catastrophic forgetting. There is a significant body of work for online learning methods with delayed labels with which the methods in the paper should be compared with.

The paper first describe how different ideas do not work and then proposes a simple technique that works better. The first part of the paper is of limited interest. It would be more useful if the authors compare the technique they propose with other methods for online learning with delayed labels.

**Questions:**

Why you describe your methods as "continual learning"? It seems that the performance over past data is not evaluated so there cannot be any forgetting

Typos such as “ data. and deployed” should be avoided

---

> ### Author Response · Authors · 2023-11-22
> **Reply to reviewer Leqv**
>
> We agree with the reviewer that we need to differentiate our contributions from the works in Online Learning as we propose a Continual Learning framework. To this end we provide an extensive comparison on forgetting in [F] (see resources in the general comment). We exclude TTA from the evaluation as the feature representation is not altered by the training procedure in our TTA adaptation. Furthermore, note that on CGLM our method achieves twice the amount of backward transfer than the relevant baselines.
>
> Online Learning and Online Continual Learning, while both involve learning from data arriving sequentially, differ fundamentally in scope. Online Learning typically deals with single-task streams, often assumed to be from an i.i.d. distribution, as outlined in section 2.3 of [20] and the introduction of [21]. In contrast, Online Continual Learning (OCL) is more concerned with non-stationary streams that undergo frequent changes in distribution, where mitigating forgetting is one of several challenges [21, 18, 22].
>
> Regarding our selection of metrics, we note that recent work in OCL, especially in scenarios with task-free or unclear task boundaries, has shifted the emphasis towards reporting final accuracy [22, 23] and adaptation to future samples (or domains) [14, 24, 16], rather than focusing on the measurement of forgetting (i.e., backward transfer). Consistent with [16], our work similarly emphasizes adaptation to future samples as a primary metric. However, for a comprehensive assessment, we will include backward transfer results in the appendix.

---

### Official Review · Reviewer_jtG8 · 2023-10-30

**Soundness:** 3 good
**Presentation:** 3 good
**Contribution:** 2 fair
**Rating:** 3
**Confidence:** 4

**Summary:**

This paper considers the problem of continuous learning with label delay, where the ground truth label of certain data in the stream is available only after certain timestamps. The authors validate the negative effect of label delay on several recent continuous image classification datasets and provide an importance sampling based method to resist the label delay.

**Strengths:**

This work considers an important problem in continuous learning, and the empirical results on the recent large-scale image dataset provide good insights for this line of research.

**Weaknesses:**

1. There exist overclaim on the contribution of this paper. The problem setting of continuous learning or online learning with label delay is observed and studied over a decades ago and there are many both theoretical and empirical studies on this problem, please see [1,2,3,4] and reference in these papers. There is lack of discussion and comparison with this line of research.


[1] Weinberger, Marcelo J. and Ordentlich, Erik. On delayed prediction of individual sequences. IEEE Transactions on Information Theory, 2002.

[2] Kuncheva LI, Sánchez JS. Nearest neighbour classifiers for streaming data with delayed labelling. In: ICDM, 2008.

[3] Quanrud, Kent, and Daniel Khashabi. Online learning with adversarial delays. In: NIPS, 2015.

[4] Flaspohler G E, Orabona F, Cohen J, et al. Online learning with optimism and delay. In: ICML, 2021.


2. Results in section 4 provides a good validation on the recent images dataset in continuous learning that the multiple gradient descents in historical data may not benefit generliazation on newly arrived unlabled data due to distribution change. However, in my opinion, it is overclaiming to be a contribution as it is a known result. If there are not enough data at each time stamp (typically in continuous learning setting) and lack of well-designed regulazation term, the performance will even degrade with increasing number of gradient descent because the model overfits to such limited number of previous data with different distribution.

3. Experimental comparisons are unfair. Self-supervised learning and test-time adaptation methods are designed for the learning scenario where the learner does not have access to historical labeled data, or in some test-time training algorithms, the learner can only modify the training process in the training time. Comparing these algorithms with importance sampling on historical data is unfair, since in the latter case we can obtain and use the ground truth label of the historical data. It is more reasonable to consider rehearsal-based continuous learning algorithms as a comparison.

**Questions:**

What is the performance of the proposed importance sampling method compared to rehearsal based continuous learning algorithms in the setting of continuous learning with label delay?

---

> ### Author Response · Authors · 2023-11-22
> **Reply to reviewer jtG8**
>
> 1. **There exist overclaim on the contribution of this paper...**
>
> The label delay and the literature the reviewer is referring to all fall in the online optimization/learning literature. While there is an overlap with our work that is focused on online continual learning, the setup here is far more challenging. Please read our response in the general comments above on the differences between the 2 domains both in the definition of the problems and the algorithms used. As we explain in detail in [F] existing work does not overlap or does not scale to this regime. More specifically, two of the four suggested works do not even consider time-evolving distributions. The largest dataset of the 4 works [9,10,11,12] consists of 2310 data points with 19 features. Our work is the first one to consider algorithms in the large scale (39M high resolution images) setting with over 10k sample delay and the first one to evaluate backward transfer in the label delay setting.
>
> 2. **... it is overclaiming to be a contribution as it is a known result...**
>
> There is no prior art that reports this phenomena at such scale. While the observed behaviour is expected, we disagree with the reviewer that this is a known result. We do experiments on 17,000x more samples of 2,900x more dimensions than the most complex experimental setup across all suggested works - this inevitably leads to significantly different findings and contributions. Rejecting new findings on the basis of not citing outdated work puts a significant burden on scientific progress.
>
>
> 3. **Experimental comparisons are unfair.**
>
> >"Self-supervised learning and test-time adaptation methods are designed for the learning scenario where the learner does not have access to historical labeled data, or in some test-time training algorithms, the learner can only modify the training process in the training time."
>
> TTA and SSL methods do not assume access to prior data however they are particularly tailored to handle domain shift, which is our hypothesized root cause of the degradation in online accuracy. Even though they are targeting this domain shift (using the newest unlabelled data) they are incapable of bridging this gap without labels. Moreover, all TTA and SSL methods we compare against, as stated in Section 5.1 - Experimental setup, in fact are augmented with the memory replay strategy (illustrated in Figure [E]) in which these methods aim to preserve performance on prior data through memory rehearsal while adapting to new data.
>
>
> 4. **It is more reasonable to consider rehearsal-based continuous learning algorithms as a comparison.**
>
> We would like to ask for clarification, as in which methods should we directly compare against?
> More specifically, we take the best performing method from [16] namely Experience Replay (ER)[19]. In the reported experiments of [16] (e.g. Figure 3) the rehearsal based continual learning methods like MIR[17] and GSS[18], requested by the reviewer, are the *two worst performing methods* when compared against ER[19] (which is chosen to be the Naïve baseline in our paper). Since our experimental setup is identical to [16], we can directly report these results in our work. Furthermore, in Figure 3 of [16], notice that the performance gap between ER-GSS and ER-MIR in the non-delayed scenario is larger than (in Figure 1 of our paper) the gap between the non-delayed ER and the d=100 ER method.

---

### Official Review · Reviewer_xqwH · 2023-10-31

**Soundness:** 3 good
**Presentation:** 3 good
**Contribution:** 2 fair
**Rating:** 5
**Confidence:** 2

**Summary:**

This paper introduce a novel continue learning setup which includes delay labeling for data sampled from time-varying distribution. To investigate this learning scenario, It conducts experiments comparing a naive algorithm which disregards unlabeled data with self-supervised learning and Test-Time Adaption methods,  all of which prove to be inadequate for solving this challenge. Finally, the paper proposes an Importance Weighted Memory Sampling approach to match the distribution of labeled samples to the newest unlabeled samples.

**Strengths:**

The introduced continuous learning problem, incorporating label delay, reflects a real-world issue of significance. Distribution shifts and labeling delays are common challenges encountered in product cycles.

The paper is effectively motivated by highlighting the inherent difficulty of the setup and demonstrating the inadequacy of popular techniques such as Self-Supervised Learning and Test-Time Adaptation in addressing this challenge.

The proposed method, while straightforward, exhibits strong performance when compared to baseline approaches.

**Weaknesses:**

It's surprising the proposed Importance Weighted Memory Sampling (IWMS) method performs even better than the naive algorithm with no delay. IWMS aims to match the distribution of the selected samples to the newest data distribution, how could model trained on it outperform a model trained on the newest data with labels? More discussion is needed regarding its exceptional outperformance.

The paper should also include semi-supervised learning as a baseline for comparison.

**Questions:**

Can you explain why the proposed IWMS performance is so different between CLOC and CGLM? Especially, why CGLM can outperform Naive algorithm with no delay?

Some typo:
In page 4, problem setting step 5, unlabeled dataset should not include $y_i^\mathcal{T}$
In page 8, memory buffer is introduced without detailed description. What samples included in the buffer, what's the frequency to update the buffer.

---

> ### Author Response · Authors · 2023-11-22
> **Reply to reviewer xqwH**
>
> ## Strengths
>
> We thank the reviewer for their positive assessment of our work. We would like to draw the other reviewer's attention to the claims on the thoroughness of our experiment and the simplicity of our method. We believe these are indeed the key contributions of our work.
>
>
> ## Weaknesses
> 1. **how could model trained on it outperform a model trained on the newest data with labels?**
>
> It can be due to multiple factors:
> - Refer to our Ablation studies for overfitting on an outdated distribution (also mentioned by reviewer jtG8): when multiple iterations are taken, the Naive baseline might overfit on the supervised data which does not hurt the on-line accuracy when the delay is small but with increasing delay the distribution shift is increased too. This is why even RR can significantly outperform Naive (Figure 5 left in our paper).
> - In the original experiments the memory size was limited (40K), please refer to our experiments with 2^19 memory size in the updated paper. We observe that IWMS only converges to Naive w/o delay and cannot outperform it neither on CLOC nor CGLM.
>
> 2. **Can you explain why the proposed IWMS performance is so different between CLOC and CGLM?**
>
> Certainly! Our two key observations are:
> - There is little to no correlation of the image content to the label on CLOC that may lead to such extensive differences in the results, however evaluations on this dataset is expected in the Online Continual Learning literature. See the supplementary material for examples of extremely low quality images in CLOC.
> - Furthermore, the difference in performance between the two datasets is possibly due to excessive concept drift in CLOC (reinforced by findings in [15]) while close to no concept drift in CGLM. While both dataset show covariate shift, CGLM does not exhibit concept drift (input to a previously different category appears under a new label) whereas CLOC does. For example a sunny beach could belong to Australia during December, while in June a visually very similar image would be a lot more likely to be taken in Spain or the US. As pointed out by reviewer nqA5 and prior work [4], detecting and handling concept drift before observing the actual labels is theoretically impossible. While our method successfully tackles the challenges of covariate shift, in case of concept drift our method recalls samples belonging to all potential candidate classes (regardless of their future ground truth label) which may be the ideal solution to the problem without any knowledge about the future, however is less robust against unpredictable changes in the stream. Such unpredictable changes are more prominent in CLOC, thus the difference in performance.
>
> We will include this explanation in the main paper and provide details in the supplementary materials.
> However, if the reviewer is referring to semi-supervised methods that use pseudo-labeling, we have provided further  experiments in Figure [B,C,D].
>
> 3. **Typos**
>
> we thank the reviewer for the thoroughness and we have fixed the typos in the main paper
>
> 4. **In page 8, memory buffer is introduced without detailed description. What samples included in the buffer, what's the frequency to update the buffer.**
>
> The buffer follows the same First-In-First-Out implementation as it does in prior work [14,16,19] which we have already highlighted in the experimental setup, Section 4.1.

---

### Official Review · Reviewer_nqA5 · 2023-11-01

**Soundness:** 2 fair
**Presentation:** 3 good
**Contribution:** 2 fair
**Rating:** 5
**Confidence:** 3

**Summary:**

The paper studies the problem of online learning, where the goal is to maintain a machine learning model in a data stream. They study this problem with the additional challenge of label delay, which occurs when the label of a data point is delayed and hence it cannot be immediately used for training.

They run different experiments using existing techniques such as SSL (Semi-Supervised Learning) and TTA  (Test-Time adaptation) to leverage the unlabeled data, however, in their analysis, they find out that these methods do not outperform the naive approach to discard the unlabeled data until one receive the labels. To address this problem, they propose a new method called Weighted Memory Sampling that leverages the unlabeled data as follows: the unlabeled data is used to find labeled data points that are similar (feature representation) so that we can use the latter data for training. This heuristic allows the authors to address distribution drift since it can possibly adapt to the change in distributions by using similar points from the past.

They run different experiments to verify the effectiveness of their approach and show that it indeed performs better than the previously tried baselines.

**Strengths:**

The paper addresses an extremely important problem, and label delay is a relevant challenge in most of practical applications with data streams.

The strength of the paper lies in the empirical evaluation, as they convincingly try different baselines and their method using a real dataset. They try different combinations of hyper-parameters (delay, computational budget) and thoroughly discuss the obtained results.

I believe that the method introduced by the authors (Weighted Memory Sampling) leverages in a very clever way the information of the unlabeled data.

I have a couple of suggestions:
- the paper [Yao, Huaxiu, et al. "Wild-time: A benchmark of in-the-wild distribution shift over time." NeurIPS (2022)] provides other additional benchmarks that could be used.
- It would be interesting to relate your measure of similarities with other quantities that were computed in domain adaptation (e.g., discrepancy) that also try to leverage unlabeled data, albeit in a different setting (as[ Mansour, Yishay, Mehryar Mohri, and Afshin Rostamizadeh. "Domain Adaptation: Learning Bounds and Algorithms.]"). In particular, I think your method can only work correctly if there is a shift in the covariate distribution, but it cannot detect a shift in the label distribution (but this is probably impossible because of the label delay).

**Weaknesses:**

I think that the critical weakness of the paper is that it does not provide a convincing comparison of its contribution with respect to existing work. There is no discussion of existing work on label delay.

However, the problem of label delay has definitely already been studied in the literature. I think that the statement that no prior work addressed this problem is too strong, and a (big) section of related work should be dedicated to those existing works.

Just a few papers, but there are many more on this topic:
[A]: Mesterharm, Chris. "On-line learning with delayed label feedback." International Conference on Algorithmic Learning Theory. Berlin, Heidelberg: Springer Berlin Heidelberg, 2005.
[B]: Gomes, Heitor Murilo, et al. "A survey on semi-supervised learning for delayed partially labelled data streams." ACM Computing Surveys 55.4 (2022): 1-42.
[C]: Gao, Haoran, and Zhijun Ding. "A Novel Machine Learning Method for Delayed Labels." 2022 IEEE International Conference on Networking, Sensing and Control (ICNSC). IEEE, 2022.
[D]: Plasse, Joshua, and Niall Adams. "Handling delayed labels in temporally evolving data streams." 2016 IEEE International Conference on Big Data (Big Data). IEEE, 2016.
[E]: Hu, Hanqing, and Mehmed Kantardzic. "Sliding Reservoir Approach for Delayed Labeling in Streaming Data Classification." (2017).
[F]: Souza, Vinicius MA, et al. "Classification of evolving data streams with infinitely delayed labels." 2015 IEEE 14th International Conference on Machine Learning and Applications (ICMLA). IEEE, 2015.

**Questions:**

- How does your paper stand with respect to existing work on label delay? (See Weakness)?

---

> ### Author Response · Authors · 2023-11-22
> **Reply 1/2 to reviewer nqA5**
>
> We thank the reviewer for recognizing our research direction, extensive experiments, and proposed alogrithm, as well as their valuable suggestions. Here are our responses to each of the comments.
>
> 1.  **additional benchmarks that could be used**
>
> The suggestion for the benchmark is a really good idea, and the proposed dataset Wild-time contains a large number of of real world problems, which is quite aligned with our original intention of the paper. However, we want to clarify some properties of our online continual learning setting. Our experimental stream is ordered strictly over time, and each sample on the stream has a unique time step label.
> On such a stream, the batch is for the evaluation of the last time step and for training of the current time step with whatever choice of batch size, there is always the time shift of one batch.
> Most tasks in the Wild-Time benchmark do not have such properties, and thus require further considerations and modifications. Due to the time limit, we could not evaluate our approach on Wild-Time. We will cite them and benchmark Wild-Time in our future research.
>
> 2.  **It would be interesting to relate your measure of similarities with other quantities that were computed in domain adaptation**
>
> We thank the reviewer for mentioning the distribution shift. Measuring the distribution shift explicitly is an interesting and important avenue for the further analysis of our method. However, some traditional domain adaptation metrics, such as Maximum Mean Discrepancy (MMD), $\mathcal{H}$ distance, or Wasserstein distance, are hard to measure in an ever-changing online stream. Most of them rely on two distinct, time-invariant disitributions (namely source and target) with large amounts of data readily available. They would be highly sensitive to the design choices of model and data.
>
> We want to note that such analysis has been already carried out on the two benchmarks we use in prior literature, and is tightly correlated with the quantities suggested by the reviewer for measuring the shift:
> - In CLOC [14] there is a section dedicated to "Validating the Distribution Shift of CLOC" where they compare the performance of two models, one trained on a limited temporal range, and the other trained on the entire temporal range, over the validation stream. The degradation of the first mentioned model demonstrates the distribution shift.
> - RapidOCL [15] uses "blind classifiers" predictors that completely ignore the input data yet still achieve unrealistically high accuracy due to strong temporal label correlations.
>
> 3. **your method can only work correctly if there is a shift in the covariate distribution, but it cannot detect a shift in the label distribution**
>
> Your description is indeed accurate. Our proposed method IWMS is, indeed, designed  for covariate shifts, since we hope to synthesize future batches by data from previous time-steps.
>
> Actually dealing with label drift(also known as concept drift, where the input data distribution remains the same/similar but the corresponding label distribution changes) is intrincally hard and almost impossible, as mentioned in [4]: "One interesting connection between semi-supervised learning and unsupervised concept (label) drift detection is that if the underlying marginal data distribution 𝑃 (𝑋) over the input does not contain information about 𝑃 (𝑦|𝑋) or indicates changes on 𝑃 (𝑦|𝑋), then it is impossible to exploit unlabelled data to improve a supervised learner (SSL) or detect a change."
>
> However, in our framework, if the label shift between two categories (either A->B or B-A) has been observed in the past, IWMS will recall samples belonging to both categories that can help the feature extractor to further refine the discriminative patterns used for distinguishing the two categories.  To illustrate this consider the following example: if the model has seen sandy beach images before from Australia (A) and later from Brazil (B), then later on when the stream reveals a new image of a similar sandy beach then the optimal policy is to recall both (A) and (B) from the memory and fine-tune the model to learn more discriminative features on the pictures. This in turn will result in increasing the chances of the classifier making the correct prediction on the yet to be labeled image of a sandy beach.

---

> ### Author Response · Authors · 2023-11-22
> **Reply 2/2 to reviewer nqA5**
>
> 4. **it does not provide a convincing comparison of its contribution with respect to existing work.**
>
> We thank the reviewer for bringing up a list of literature. We find the shared work extremely insightful. However, we  argue that the majority of prior art referenced by the reviewer (which we commend) is in fact within the online learning (from optimization perspective) literature as opposed to *continual learning* online/offline literature that is a concern of the paper. The main issue seems to be the contextualization of our work.
>
> Among them, [4] and [8] are the most related to our work. However, they cannot compete in our limited computational budget as highlighted by the paper [4]: “the main challenge in adopting such a strategy to a streaming [online] scenario is that it requires multiple passes over the input data”.
>
> Pseudo labeling is one important perspective of [B], and we had already done the experiments of using pseudo labeling. The experiments failed since
> - The base classifier C has really low accuracy from the beginning, the quality of the pseudo labels were extremely poor, always leading to a feedback loop on training wrong labels.
> - The correct predictions often have low confidence scores (top-1 softmax scores), and it makes the pseudo-label selection, proposed in [B], unstable. Furthermore, in the attached Figure [C,D], by increasing the delay it becomes less and less probable that a correct prediction will have a high enough confidence score
>
> We have a more detailed description of each of the literatures in our provided link [G]. Nevertheless, we will cite the ones that address our problem in particular. If the reviewer considers this a relevant baseline to compare against, we are happy to include the full results in the updated version of the paper.

---

> > ### Comment · Reviewer_nqA5 · 2023-12-04
> >
> > I thank the authors for their detailed response. After reading the rebuttal and the other reviews, I decided to keep my score.
> >
> > The main reason for my score is that the paper lacks a discussion of relevant related work on label delay. Label delay has been extensively studied in the literature, as pointed out by other reviewers and discussed in detail in the rebuttal by the authors.  Although the exact setting may differ from what the authors are considering, this is still indeed relevant work that should be contextualized in the Related Work section (and arguably it should be the most important section of the related work).

---

### Author Response · Authors · 2023-11-22
**General comment**

We would like to thank the reviewers for their great suggestions. There has been a shared misunderstanding in confusing our work on Online Continual Learning with the optimization literature of Online Learning. We have not clearly stated the difference in the paper, which is something we will address in the final version. We thoroughly revisited the suggested papers and included further comparisons to reinforce our statement.

The key differences are:
- **Online Learning vs Online Continual Learning**: Online Learning and Online Continual Learning, while both involve learning from data arriving sequentially, differ fundamentally in scope. Online Learning typically deals with single-task streams, often assumed to be from an i.i.d. distribution, as outlined in section 2.3 of [20] and the introduction of [21]. In contrast, Online Continual Learning (OCL) is more concerned with non-stationary streams that undergo frequent changes in distribution, where mitigating forgetting is one of several challenges [21, 18, 22]. To highlight the difference, works suggested by reviewers nqA5 and jtG8 [2,9,11] are focusing on time-invariant distributions.

- **Non-i.i.d. distribution of unsupervised data**: none of the suggested work [1,2,3,4,5,6,7,8,9,10,11,12] differentiates between past and future unsupervised data. In our proposal, all unsupervised data is newer than the last supervised data. For clarification see Figure [A] (see resources list).

- **Considering catastrophic forgetting**: Continual Learning, both online and offline, is concerned about past data. This is different from Online learning where forgetting is not examined. To that end, it was an oversight on our part to not have reported the forgetting experiments as they have been given slightly lower priority in online continual learning works [15,16] where people have reported them only in the appendix. Nevertheless, we absolutely agree and we have now conducted the backward transfer experiments in an identical setup to [14] comparing to  3 baselines, over 6 different delay scenarios, as suggested by reviewer Leqv. We observe that our method outperforms the SSL, Naïve and non-delayed Naïve baseline achieving ~2x better accuracy on CGLM, whereas on CLOC all methods perform similarly (due to poor data quality as reported in the supplementary material of the original submission). In contrast, none of the suggested online learning literature [2,3,4,5,6,7,8,9,10,11,12] studies catastrophic forgetting.

- **Limited applicability of the suggested papers**: The majority of the suggested work has very strong limitations regarding scalability: all suggested work [2,3,4,5,6,7,8,9,10,11,12] except for [1] considers several orders of magnitude lower number of samples of tabular data. E.g. [7] considers spam detection on <10k samples using 500 bag-of-word features, while we show results on 39M high resolution images). Furthermore, methods and error bounds proposed in [3,5,6,7,9,10,11,12] do not provide significantly better results than a random classifier on our datasets due to the lack of memory rehearsal (which is necessary for forgetting, a property not of a concern in online learning). Adapting methods that are unlikely to improve on the strong baseline of ER [19] is beyond the scope of our work. On the other hand, we have adapted and evaluated 12 highly relevant methods (8 self supervised methods and 4 test time adaptation methods) to provide a strong baseline. Furthermore, following the suggestion of xqwH we have provided analysis and ablation using an additional baseline on Semi-Supervised Learning (more precisely, pseudo-labeling) in [B,C,D].

While the scope of the suggested related works is too extensive to be included in our paper, we will add as much as possible to the related works and add more detailed discussion to the supplement. We really appreciate the effort made by the reviewers, thus we provide a one-by-one comparison of each suggested paper in [G]. Furthermore, we reduced the claims in our paper from "the first method to do label delay" to "the first method to do label delay in online *continual learning* at large scale”. Moreover, the fact that label delay has been studied in a different context (online learning) in several other papers in the past highlights the importance of addressing it in the challenging large scale continual learning context.

---

> ### Author Response · Authors · 2023-11-22
> **Related Work**
>
> [1]: Yao, Huaxiu, et al. "Wild-time: A benchmark of in-the-wild distribution shift over time." NeurIPS (2022).
>
> [2]: Mansour, Yishay, Mehryar Mohri, and Afshin Rostamizadeh. "Domain Adaptation: Learning Bounds and Algorithms."
>
> [3]: Mesterharm, Chris. "On-line learning with delayed label feedback." International Conference on Algorithmic Learning Theory. Berlin, Heidelberg: Springer Berlin Heidelberg, 2005.
>
> [4]: Gomes, Heitor Murilo, et al. "A survey on semi-supervised learning for delayed partially labelled data streams." ACM Computing Surveys 55.4 (2022): 1-42.
>
> [5]: Gao, Haoran, and Zhijun Ding. "A Novel Machine Learning Method for Delayed Labels." 2022 IEEE International Conference on Networking, Sensing and Control (ICNSC). IEEE, 2022.
>
> [6]: Plasse, Joshua, and Niall Adams. "Handling delayed labels in temporally evolving data streams." 2016 IEEE International Conference on Big Data (Big Data). IEEE, 2016.
>
> [7]: Hu, Hanqing, and Mehmed Kantardzic. "Sliding Reservoir Approach for Delayed Labeling in Streaming Data Classification." (2017).
>
> [8]: Souza, Vinicius MA, et al. "Classification of evolving data streams with infinitely delayed labels." 2015 IEEE 14th International Conference on Machine Learning and Applications (ICMLA).
>
> [9]: Weinberger, Marcelo J. and Ordentlich, Erik. "On delayed prediction of individual sequences." IEEE Transactions on Information Theory, 2002.
>
> [10]: Kuncheva LI, Sánchez JS. "Nearest neighbour classifiers for streaming data with delayed labelling." ICDM, 2008.
>
> [11]: Quanrud, Kent, and Daniel Khashabi. "Online learning with adversarial delays." NIPS, 2015.
>
> [12]: Flaspohler G E, Orabona F, Cohen J, et al. "Online learning with optimism and delay." ICML, 2021.
>
> [13]: Zhang et al. "VIBE: Topic-Driven Temporal Adaptation for Twitter Classification." Proceedings of the 2023 Conference on Empirical Methods in Natural Language Processing (EMNLP).
>
> [14]: Cai, Zhipeng, Ozan Sener, and Vladlen Koltun. "Online continual learning with natural distribution shifts: An empirical study with visual data." Proceedings of the IEEE/CVF international conference on computer vision. 2021.
>
> [15]: Al Kader Hammoud, Hasan Abed, et al. "Rapid Adaptation in Online Continual Learning: Are We Evaluating It Right?." Proceedings of the IEEE/CVF International Conference on Computer Vision. 2023.
>
> [16]: Ghunaim, Yasir, et al. "Real-time evaluation in online continual learning: A new hope." Proceedings of the IEEE/CVF Conference on Computer Vision and Pattern Recognition. 2023.
>
> [17]: Rahaf Aljundi, Lucas Caccia, Eugene Belilovsky, Massimo Caccia, Laurent Charlin, and Tinne Tuytelaars. "Online continual learning with maximally interfered retrieval." Conference on Neural Information Processing Systems (NeurIPS), 2019.
>
> [18]: Rahaf Aljundi, Min Lin, Baptiste Goujaud, and Yoshua Bengio. "Gradient based sample selection for online continual learning." Conference on Neural Information Processing Systems (NeurIPS), 2019.
>
> [19]: ​​Chaudhry, Arslan, et al. "On tiny episodic memories in continual learning." arXiv preprint arXiv:1902.10486 (2019).
>
> [20]: Chen, Zhiyuan, and Bing Liu. Lifelong machine learning. Vol. 1. San Rafael: Morgan & Claypool Publishers, 2018.
>
> [21]: Fini, Enrico, et al. "Online continual learning under extreme memory constraints." Computer Vision–ECCV 2020: 16th European Conference, Glasgow, UK, August 23–28, 2020, Proceedings, Part XXVIII 16. Springer International Publishing, 2020.
>
> [22]: Bang, Jihwan, et al. "Online continual learning on a contaminated data stream with blurry task boundaries." Proceedings of the IEEE/CVF Conference on Computer Vision and Pattern Recognition. 2022.
>
> [23]: Jin, Xisen, et al. "Gradient-based editing of memory examples for online task-free continual learning." Advances in Neural Information Processing Systems 34 (2021): 29193-29205.
>
> [24]: Lin, Zhiqiu, et al. "The clear benchmark: Continual learning on real-world imagery." Thirty-fifth conference on neural information processing systems datasets and benchmarks track (round 2). 2021.

---

> > ### Author Response · Authors · 2023-11-22
> > **Supplementary material**
> >
> > [A]: [https://pasteboard.co/Jinzcr56IKEk.png](https://pasteboard.co/Jinzcr56IKEk.png) - Figure explaining the main difference between the generic partial labeling scenario and our work
> >
> > [B]: https://pasteboard.co/JqqnFUXKsv92.png - Figure showing preliminary results on pseudo-labeling (often referred to as semi-supervised learning  self-training or self-labeling in the online literature)
> >
> > [C]: https://pasteboard.co/B0LqG0R9QsnB.png - Figure showing pseudo-labeling ablation on sensitivity to varying confidence thresholds.
> >
> > [D]: https://pasteboard.co/a8JJcZxLnvZs.png - 4 histograms (for delay=0/10/50/100) illustrating the difficulty of selecting a robust confidence threshold.
> >
> > [E]: https://pasteboard.co/DCqwAXnn8KRy.png - Figure clarifying the experimental setup: in our framework the reported methods augment the Naïve method by using unsupervised data
> >
> > [F]: https://pasteboard.co/6JNDiFfGQhKu.png - Figure reporting backward transfer for measuring forgetting.
> >
> > [G]: https://justpaste.it/blrjs - Extensive one-by-one comparison with each suggested work [1-12]

---

### Meta-Review · Area_Chair_RSMF · 2023-12-09

**Metareview:**

In this paper, the authors propose a new model for delayed annotation that showed improvement compared to simple baselines, and it has a significant evaluation of the proposed method. The reviewers have significantly valued this part of the paper, and they see merit in it. However, several reviewers point out the poor literature review and the need to compare this paper with other papers. The authors suggest a significant number of papers to include in the final version of the paper in their last comment. They should have been identified earlier. Also, the reviewers point out that comparisons with state-of-the-art algorithms should have been included. Especially, reviewers nqA5 and jtG8 point out relevant comparisons. Authors should consider improving the paper following the reviewers' comments, and we encourage resubmission next year.

Finally, the mistake of uploading a copy that reveals the authors' names would be grounds for desk-rejecting the paper. If we (reviewers, AC, and SAS) had decided to accept the paper, the program chairs would have reversed the decision for violating the double-blind policy. We understand it was an honest mistake, but we take this policy seriously. We encourage you to try to double-check in future submissions.

**Justification For Why Not Higher Score:**

Paper did not do a thorough comparison and it violated double blind policy.

**Justification For Why Not Lower Score:**

there is no lower score.

---

### Decision · Program_Chairs · 2024-01-16

Reject